# The Effects of Social, Personal, and Behavioral Risk Factors and PM_2.5_ on Cardio-Metabolic Disparities in a Cohort of Community Health Center Patients

**DOI:** 10.3390/ijerph17103561

**Published:** 2020-05-19

**Authors:** Paul D. Juarez, Mohammad Tabatabai, Robert Burciaga Valdez, Darryl B. Hood, Wansoo Im, Charles Mouton, Cynthia Colen, Mohammad Z. Al-Hamdan, Patricia Matthews-Juarez, Maureen Y. Lichtveld, Daniel Sarpong, Aramandla Ramesh, Michael A. Langston, Gary L. Rogers, Charles A. Phillips, John F. Reichard, Macarius M. Donneyong, William Blot

**Affiliations:** 1Department of Family and Community Medicine, Meharry Medical College, Nashville, TN 37208, USA; wim@mmc.edu (W.I.); pmatthews-juarez@mmc.edu (P.M.-J.); 2School of Graduate Studies and Research, Meharry Medical College, Nashville, TN 37208, USA; mtabatabai@mmc.edu; 3RWJF Professor, Department of Family & Community Medicine AND Economics, University of New Mexico, Albuquerque, NM 87131, USA; rovaldez@aol.com; 4Department of Environmental Health Sciences, College of Public Health, Ohio State University, Columbus, OH 43210, USA; dhood@cph.osu.edu; 5Department of Family Medicine, University of Texas Medical Branch, Galveston, TX 77555, USA; cpmouton@UTMB.edu; 6Department of Sociology, Ohio State University, Columbus, OH 43210, USA; colen.3@osu.edu; 7Universities Space Research Association, NASA Marshall Space Flight Center, Huntsville, AL 35805, USA; mohammad.alhamdan@nasa.gov; 8Department of Environmental Health Sciences, Tulane University School of Public Health and Tropical Medicine, New Orleans, LA 70112, USA; mlichtve@tulane.edu; 9Department of Biostatistics, Xavier University, Cincinnati, OH 45207, USA; dsarpong@xula.edu; 10Department of Biochemistry, Cancer Biology, Neuroscience & Pharmacology, Meharry Medical College, Nashville, TN 37208, USA; aramesh@mmc.edu; 11Department of Electrical Engineering and Computer Science, University of Tennessee, Knoxville, TN 37996, USA; Langston@eecs.utk.edu (M.A.L.); cphill25@eecs.utk.edu (C.A.P.); 12National Institute for Computational Sciences, University of Tennessee, Knoxville, TN 37996, USA; grogers3@utk.edu; 13Department of Environmental Health, Risk Science Center, University of Cincinnati, Cincinnati, OH 45221, USA; reichajf@uc.edu; 14Division of Outcomes and Translational Sciences, College of Pharmacy, Ohio State University, Columbus, OH 43210, USA; donneyong.1@osu.edu; 15Center for Population-based Research, Vanderbilt University, Nashville, TN 37235, USA; william.j.blot@Vanderbilt.Edu

**Keywords:** cardio-metabolic disease, PM_2.5_, cardiovascular disease, diabetes, stroke, personal, clinical and environmental risk factors, health disparities

## Abstract

(1) Background: Cardio-metabolic diseases (CMD), including cardiovascular disease, stroke, and diabetes, have numerous common individual and environmental risk factors. Yet, few studies to date have considered how these multiple risk factors together affect CMD disparities between Blacks and Whites. (2) Methods: We linked daily fine particulate matter (PM_2.5_) measures with survey responses of participants in the Southern Community Cohort Study (SCCS). Generalized linear mixed modeling (GLMM) was used to estimate the relationship between CMD risk and social-demographic characteristics, behavioral and personal risk factors, and exposure levels of PM_2.5_. (3) Results: The study resulted in four key findings: (1) PM_2.5_ concentration level was significantly associated with reported CMD, with risk rising by 2.6% for each µg/m^3^ increase in PM_2.5_; (2) race did not predict CMD risk when clinical, lifestyle, and environmental risk factors were accounted for; (3) a significant variation of CMD risk was found among participants across states; and (4) multiple personal, clinical, and social-demographic and environmental risk factors played a role in predicting CMD occurrence. (4) Conclusions: Disparities in CMD risk among low social status populations reflect the complex interactions of exposures and cumulative risks for CMD contributed by different personal and environmental factors from natural, built, and social environments.

## 1. Introduction

Cardiovascular disease (CVD), stroke, and diabetes comprise a spectrum of related cardio-metabolic disease (CMD) conditions: CVD (e.g., coronary heart disease), congestive heart failure, myocardial infarction, atrial fibrillation, vascular endothelial dysfunction, stroke and atherosclerosis, and diabetes (type 1, type 2, and metabolic syndrome) [1,2,3]. Together, they are the first, fifth, and seventh leading causes of death, respectively, for both men and women in the United States [4].

These diseases taken together often are referred to as CMD due to common individual risk factors (behaviors, personal traits, cardio-metabolic characteristics) as well as shared, external environmental (natural, built, and social) exposures [5]. Common individual-level CMD risk factors include some that are fixed or non-modifiable, and unique to individuals (e.g., age, race, gender) [6], some that are modifiable (diet, exercise, and smoking), and others that are metabolic. Metabolic risk factors for CMD include hypertension [7], allostatic load [8,9,10], dyslipidemia [11], decreased high density lipoprotein (HDL) and increased low density lipoprotein (LDL) cholesterol [12,13], triglycerides [14], fasting insulin [15], serum creatinine [16], serum uric acid [17] serum high-sensitivity C-reactive protein (hsCRP) [18], inflammation [19], hypertriglyceridemia [20], thrombosis [21,22], insulin resistance [23], serum lipids [24] and blood glucose [25,26], fibrinogen [27], and homocysteine [28] (see Table 1 for an extensive list of individual risk factors for CMD).

Compared with individuals without diabetes, patients with type 2 diabetes mellitus (TTDM) are one and a half times more likely to have a stroke and two to four times more likely to die from heart disease. In contrast to CVD and stroke, which have declined in recent years, the incidence and prevalence of diabetes doubled nationally between 1980 and 2008, before plateauing between 2008 and 2012. Between 2012 and 2015, the incidence of diagnosed diabetes among adults aged 18 and over decreased, while the prevalence has continued to increase [29]. Increases in both the prevalence and incidence of diabetes among subgroups, however, have continued, including for non-Hispanic Black and Hispanic subpopulations and those with a high school education or less [30].

Environmental disparities in CMD outcomes have been found to be associated with exposures to chemical and non-chemical stressors found in the natural, built, and social environments. Exposures to toxicants in the natural environment linked to CMD outcomes include heavy metals (lead, mercury, cadmium, and arsenic), solvents, pesticides, indoor pollution (secondhand smoke, biomass fuels), outdoor air pollution comprised of complex mixtures of gases that include particulate matter (PM), which includes PM_10_ (course), PM_2.5_ (fine) and ultrafine PM, carbon monoxide (CO), ozone (O_3_), nitrogen dioxide (NO_2_), sulfur dioxide (SO_2_), diesel and other sources (see Table 1). PM is a mixture of solid and liquid droplets present in the air that vary in mass, number, size, shape, surface area, chemical composition as well as reactivity, acidity, solubility, and origin [31].

Numerous epidemiological studies have found a strong association between ambient PM (PM_10_, PM_2.5_, and ultrafine particles) and increased CMD, including myocardial infarction (MI) [32,33], cardiac arrhythmias [34,35], vascular dysfunction [36,37], hypertension [38,39], diabetes [40,41], ischemic stroke [42,43], and atherosclerosis [44,45], even at relatively low concentrations. Risk factors in the built environment that have been found to be related to CMD include interaction with nature (green space, walkability, activity supportive built environment, and lack of places to exercise) [46,47] and access to a healthy food environment (distance, affordability, and availability of healthy food and high-density of fast-food restaurants) [48] (see Table 1). Social factors associated with CMD outcomes include race/ethnicity [49], neighborhood deprivation [50], safety (neighborhood violent crime and unemployment) [51], social capital [52], and rural vs. urban [53]. Other social risk factors that have been found to be associated with CMD include population density, community stressors, residential segregation, health insurance, access to health care services, lack of trust in health care providers, socio-cultural beliefs and norms (car ownership, cultural influences), and availability of social supports (see Table 1). Public policies that have been identified as risk and protective factors for CMD disparities are local laws and regulations that have a direct or indirect outcome on CMD, including increased access to healthcare through Medicaid expansion under the Affordable Care Act, restrictions on cigarette smoking, zoning ordinances regarding parks, walking and biking paths and policies that encourage use of public transit (see Table 1).

Several studies found Black:White racial disparities in the association between exposure to PM_2.5_ and cardio-metabolic outcomes. The Multi-Ethnic Study of Arherosclerosis (MESA) study [54] found that Blacks compared to Whites, showed a stronger adjusted association between air pollution and left-ventricular mass index (LVMI) and left-ventricular ejection fraction (LVEF). The MESA study also found that higher exposure to multiple chemical constituents of air pollution may be a novel contributor to diabetes disparities [55]. Data from the HeartSCORE study found significant Black:White racial disparities between exposure to PM_2.5_ and higher blood glucose, worse arterial endothelial function, and incident CVD events [56].

The present study examined the effects of PM_2.5_ exposure on Black:White disparities in CMD by linking daily measures of PM_2.5_ with survey responses of individual and environmental risk factors and CMD history taken at enrollment of participants of the Southern Community Cohort Study (SCCS) [57].

## 2. Materials and Methods

### 2.1. Aim

The overall aim of this study is to assess self-reported clinical, personal, and environmental risk factors, measures of PM_2.5_, and risk for Black:White CMD disparities among participants of the SCCS.

### 2.2. Design

The cross-sectional study design combined individual SCCS participant survey responses at enrollment with an annualized daily measure of PM_2.5_ for the 12-month period prior to study enrollment.

### 2.3. Sample

The initial SCCS cohort consisted of 84,513 individuals who were recruited from March 2002–September 2009 in 12 southeastern states (Alabama, Arkansas, Florida, Georgia, Kentucky, Louisiana, Mississippi, North Carolina, South Carolina, Tennessee, Virginia, West Virginia) [55,56]. Individuals under treatment for cancer within a year of enrollment were excluded from the original study. Participants were recruited primarily (86%) from Community Health Centers (CHCs) [57] where trained interviewers collected survey information on illness history, lifestyle, social-demographic, and environmental factors. A smaller percentage of cohort participants were enrolled using a mailed questionnaire sent to a stratified random sample of residents in the same states.

For the current study, participants were limited to those for whom 12 months of daily PM_2.5_ data at date of enrollment were available (*n* = 72,215). Only persons who identified as either White or Black and who had complete data for variables included in the model were included (*n* = 53,617). Persons who were not recruited through community health centers were excluded from the study. The final sample thus was comprised of those persons for whom we had 12 months of daily PM_2.5_ data, were either Black or White, had no missing data of those variables that were used in the model and were recruited through community health centers (*n* = 48,799). The current research protocol was approved by the Meharry Medical College Institutional Review Board (IRB) and the EPA Human Subjects Research Review Official, while permission for data access was provided by the SCCS governing board. The Meharry and Vanderbilt University IRBs approved and oversaw SCCS recruitment. All SCCS participants provided written informed consent.

Ethics Approval and Consent to Participate: De-identified secondary data on individual health records were used. There was no intervention or direct interaction with human subjects. The end points of the research are to identify and model the mechanisms and exposure pathways associated with CVD and other chronic diseases. The study protocol was approved by the Meharry Medical College (IRB Protocol # 17-11-783, Juarez) and by the EPA Human Subjects Research Review Official (HSR-000867).

### 2.4. Procedures

The following definitions were used. 

Cardio-metabolic disease (CMD): The presence of CMD was defined for SCCS participants who responded to the survey administered at the time of the enrollment by a statement that a “doctor has told you that you have” one or more of the following cardio-metabolic diseases: diabetes, heart attack or coronary artery bypass surgery, or stroke.

Fine particulate matter (PM_2.5_): PM_2.5_ exposure was defined as the average annual concentration (µg/m^3^) calculated using a continuous, spatial surface model created by Al-Hamdan et al. [102] that merged ground level ambient air measures with satellite-derived daily measures of PM_2.5_. Satellite measures were derived from regression models of aerosol optical depth (e.g., the measure of the degree to which sunlight is scattered and absorbed by aerosols of various sizes through the entire atmospheric column) collected by the Moderate Resolution Imaging Spectro-radiometer instrument onboard the National Aeronautics and Space Administration Aqua satellite (see [102] for more detailed information). An example of a map of annualized data is presented in Figure 1. A B-spline smoothing algorithm was used to calculate daily concentrations of PM_2.5_ for each 3-km grid cell for the 12 months prior to enrollment. Satellite data were used to fill the temporal and spatial gaps inherent to ground-level monitoring station data which mostly are collected from urban areas. 

Geographic identifiers and residential address proxy: Geocoded individual residential addresses at time of enrollment were assigned to PM_2.5_, 3-km grid cells. These residential address proxy grid cells were used to link survey and environmental data in order to provide a firewall between study and SCCS data and ensure personal identification and health information remained anonymous.

Clinical risk factors: Participants were asked about the presence of clinical risk factors for CMD at enrollment, by responding yes or no to the following questions: “Has a doctor told you that you have had” high blood pressure (hypertension)? or high cholesterol? Other clinical risk factors were assessed by enrollment interviewers including height, weight, and body mass index (BMI) defined as weight (kg)/height (m)^2^, the latter of which is a commonly used metric to assess obesity (with categories of <18.5 = 1, 18.5–24 = 2, 25–29 = 3, 30–34 = 4, 35–39 = 5, 40 and higher = 6, respectively designated as underweight, normal weight, overweight, obesity I, obesity II, and obesity III).

Personal risk factors Self-reported personal risk factors obtained from survey responses at enrollment included: age (45–64 = 1, 65 and older = 2; age at enrollment in years was modeled); history of tobacco use/smoking status (non-smoker = 1, former smoker = 2, smoker = 3); air quality outdoors (1 = poor, 2 = fair, 3 = good, 4 = excellent); air quality indoors (1 = poor, 2 = fair, 3 = good, 4 = excellent); educational level (1 = <9 years, 2 = 9–11 years, 3 = high school or GED, 4 = vocational training, 5 = some College, 6 = college, 7 = Masters, 8 = Doctorate); household income (1 = <$15,000, 2 = $15,000–$24,999, 3 = $25,000–$49,999, 4 = $50,000–$99,999, 5 = >$100,000); marital status (married or with a partner = 1, divorced = 2, widowed = 3, single = 4); employment status (employed = 1, otherwise 0); gender (male = 1, female = 0); residence location (rural/farm = 1, urban = 0); and race (White = 0, Black = 1).

### 2.5. Statistical Analysis

Frequency distributions of participant characteristics were tabulated for the analytic sample. Cross-tabulations of categorical variables associated with CMD were evaluated using Chi-squared tests. Sample characteristics and percentages reporting CMD for each clinical and personal risk factor were characterized. IBM SPSS 26 (IBM Corp., Released 2019. Armonk, NY: IBM Corp.) was used to perform statistical analysis and Mathematica software was used to draw graphs.

Generalized linear mixed modeling (GLMM) was used to estimate the relationship between CMD and social-demographic characteristics, behavioral and environmental risk factors, and exposure levels of PM_2.5_ [103,104]. GLMM used fixed effects (age, sex, race, etc.) and a random intercept model with these data to account for clustering of observations by state, a design feature of the SCCS. A random intercept was used to take into consideration state-dependent CMD variation. State-level factors often affect measurements similarly for any given participant in a particular state. For example, each state pursues different environmental protection laws and local zoning ordinances.

To account for state-dependent CMD variation, we modeled the random effects for the intercept using state of residence with fixed effects for individual risk factor characteristics (age, sex, smoking, etc.) and PM_2.5_ exposure of the participants. To account for the variety of possible variance-covariance structures in the relationships among SCCS participants, we used variance components structure. This is a natural way to represent participants within a state cluster. If the true correlation structure is compound symmetry, then using a random intercept for each state will remove the correlation among the participants.

A logit link function was used to model fixed effects of the presence of CMD as a binary outcome variable. The fixed effects for a GLMM are interpreted in the same way as a regression analysis depending on the nature of the outcome variable. In this case, we interpreted the model as we would a logistic regression model. The parameter estimates given in Table 1 were estimates of the mean parameters. Estimates of covariance parameters were used to identify the variance parameter, the random intercept for each state. We used a variance component structure for this parameter with a variance of zero, such that the null hypothesis would indicate that a random effect was not needed. This was tested using the Wald Z statistic.

## 3. Results

Frequencies and percentages of the participants reporting CMD by personal, health, social-demographic, and environmental characteristics are presented in Table 2. The SCCS CHC cohort sample was comprised of 60.4% females and 39.6% males. The majority of respondents were Black (66.1%) and the remaining (33.9%) were non-Hispanic White. The average age at enrollment was 52 years old. Most participants at baseline were under 65 years of age (89.7%).

Overall, 29.2% of participants responded that their primary care doctor had previously told them they had a cardio-metabolic disease. The CMD prevalence was higher among seniors (45.2%) than among those younger than 65 (27.3%) but did not differ greatly between Blacks (28.9%) and Whites (29.7%) (*p* = 0.072). Black women reported more CMD than White women, but Black men reported less CMD than White men.

CMD prevalence was inversely related to education and family income levels. Participants with lower levels of education reported higher rates of CMD, ranging from 41.7% for all subjects with less than a high school education to 19.5% among subjects with a doctorate level of education (*p* < 0.001). Similarly, respondents who reported lower family incomes reported higher rates of CMD, ranging from 32.1% for those earning less than $15,000, to 14.5% for those with family incomes of $100,000 or more (*p* < 0.001).

Participants with clinical risk factors (i.e., hypercholesterolemia, hypertension) reported higher than average levels of CMD. In addition, CMD prevalence monotonically increased with rising BMI, with 17% of those of normal weight vs. 45% of those in obesity class III reporting CMD. In these univariate data, a low, rather than high, prevalence of CMD was seen among current smokers compared with former or never smokers.

SCCS respondents who reported poorer outdoor (*p* = 0.001) and indoor (*p* = 0.105) air quality also reported higher rates of CMD. Respondents who lived in rural areas reported higher prevalence (32.5%) of CMD than those who did not (26.4%) (*p* < 0.001). CMD prevalence was 29.6%, 29.0%, 29.3%, 31.2%, 28.1%, and 25.5% within the 10th, 25th, 50th, 75th, and 90th percentiles of PM_2.5_ concentrations. The average annual PM_2.5_ concentration for individuals in the 12-state sample was 13.5 μg/m^3^ for respondents in the 12-month period prior to enrollment, with 10th, 25th, 50th, 75th, and 90th percentile values of 11.3 μg/m^3^, 12.4 μg/m^3^, 13.5 μg/m^3^, 15.0 μg/m^3^, and 15.8 μg/m^3^, respectively. The annual mean PM_2.5_ was above the 2012 EPA regulation which defined a three-year, average annual mean exposure above 12.0 μg/m^3^ as harmful to public health and the environment. Among respondents, 84.1% were exposed to PM_2.5_ above 12.0 μg/m^3^ in the 12-month period prior to study enrollment.

Table 3 shows results from GLMM multivariate modeling of the CMD in relation to the risk factor data shown in Table 2. The univariate associations seen in Table 2 tended to persist after simultaneous adjustment for other factors, with strong increases in the odds of having CMD associated with lower levels of education and income, hypertension, hypercholesterolemia, and an increasing rate of BMI. However, the significantly lower risk among current vs. never smokers and higher risk in rural vs. urban areas seen in Table 2, no longer held in the adjusted analyses.

When PM_2.5_ was modeled as a linear variable, we found higher occurrences of CMD among participants with a higher level of exposure to PM_2.5_. Exposure to increasing levels of PM_2.5_ (F = 7.569; *p* = 0.006) as measured from the algorithms developed by Al-Hamdan et al. [104] was associated with increasing prevalence of CMD for participants in the SCCS. Our findings indicate that each unit increase in PM_2.5_ is associated with an approximately 2.6% increase in the log odds of CMD for the study sample of SCCS participants.

We illustrate the findings from our model by presenting the predicted probability of CMD for four “typical” types of participants as PM_2.5_ concentrations rise. Figure 2 shows how the presence of hypertension, alone, affects the probability of CMD by race and gender at different levels of PM_2.5_ exposure. This race × gender exemplar controls for other common CMD risk factors. It assumes an individual is 52 years old, married, employed, never smoked, lives in an urban area, possesses a high school diploma, has an average weight (BMI 25–30), and lives in a home where they indicate indoor and outdoor air quality as good (i.e., a person with low CMD risk factors). In contrast, Figure 3 illustrates how having two clinical factors, hypertension and being obese (BMI 30–35), while being exposed to rising levels of PM_2.5_, affects the predicted probability of CMD. In both cases, the presence of clinical risk factors (i.e., hypertension alone, and hypertension and obesity (high BMI)) increases the predicted probability of CMD as PM_2.5_ concentrations increase.

We tested the GLMM using a classification table and confirmed that our model correctly predicted the presence of CMD 74.8% of the time (see Table 3). The intercept variance = 0.017 (Wald Z = 1.98, *p* = 0.047) (see Table 4). The null hypothesis for this parameter is a variance of zero, which would indicate that a random effect is not needed. Thus, we rejected the null hypotheses concluding a random intercept for each state is needed. These results further suggest that there are important unmeasured explanatory variables for our participants at the state level that affect CMD in a way that appears random because we do not know the value(s) of the missing explanatory variable(s).

## 4. Discussion

Our study results in four key findings:(1)Residents of communities with exposure to higher levels of PM_2.5_ annual concentrations are more likely to have reported a CMD.(2)Race, a social risk factor for disparities in health, is not predictive of CMD when behavioral, clinical, and environmental risk factors are accounted for in the model. Similarly, residence in an urban or rural setting is not associated with CMD after PM_2.5_ and other risk factor information are taken into consideration.(3)A significant residual variation in the presence of CMD among participants across states was found, perhaps reflecting differences in environmental exposures, social policies, and other place-based factors. These differences will be explored further in future analyses.(4)Multiple individual and environmental risk factors are associated with self-reported CMD, consistent with a multifactorial etiology of these conditions. Our results are generally consistent with previously published literature. We found statistically significant positive associations between CMD and marital status, BMI, education, gender, age, employment, and higher concentrations of PM_2.5_.

However, when we controlled for other clinical and social characteristics within our study population of low socioeconomic status (SES) individuals, there were no statistically significant associations between the presence of CMD and race or residence in urban vs. rural settings. In some previous studies, race has been identified as both a biological [105] and social [106] risk factor in health disparity studies; our finding demonstrates that when clinical, behavioral, and environmental factors are included in a model ascertaining the presence of CMD, race, as a social construct, is not a statistically significant predictor of CMD risk among Black and White populations compared within the same social strata. This confirms results reported by LaVeist et al. [107] who similarly found that when comparing Blacks and Whites who live in neighborhoods with similar levels of social and economic resources/exposures, the influence of race in health outcomes largely disappeared.

Despite the completion of the human genome in 2003, and the increased attention and resources given by academicians and the NIH in support of research to identify genetic differences underlying racial health disparities, there is considerable evidence that some CMD risk factors are substantially influenced by environmental factors [108]. Our findings suggest that a more holistic, exposomics approach that incorporates a broad range of exogenous and endogenous environmental risk factors experienced over the course of life may be needed to explain CMD racial disparities.

Our findings support the notion that low-income persons, regardless of race, are exposed to social disadvantage and adverse environmental exposures resulting in intransigent inequities [109]. The underlying causes of CMD are likely a result of pathophysiological insults that occur in response to biological and personal risk factors and adverse exposures associated with the natural, built, and social environments. Extrapolating our regional findings within the SCCS, it also seems likely that racial disparities in CMD observed nationally are mainly due to differences in these multiple risk factors, rather than intrinsic to race per se. This finding has implications both for CMD research, the science of health disparities, and for the conceptualization of interventions in the clinical, public health, and policy arenas [110].

## 5. Conclusions

The nature of the SCCS cohort, which is overwhelmingly of low socio-economic status (SES) and within the southeastern region of the US, where CVD, stroke, and diabetes prevalence are high, allows us to compare Black and White subgroups whose disease “riskscapes” are more similar than different. Our results support the finding that race, as a risk factor for disease, as well as a way to elaborate on the patterning of CMD, should not be interpreted as “biological pre-programming”. Rather, our results indicate that CMD disparities reflect the complex interactions of personal risk factors and exposures that emerge from the social, built, and natural environments among low social status populations. Our results echo those of Geronimus and colleagues [111], who demonstrated similar mortality risks among high-poverty Black and White rural populations. Our findings underscore that position within social strata indeed matters with regards to health outcomes and the need for additional studies that critically examine racial health disparities within SES subgroups or social strata, particularly those in different geographic areas.

Modeling the contributions of multiple environmental exposures on CMD health and health disparities, as experienced in the real world, is in its formative stage. The “exposome framework” presents a new approach for considering the complex ways in which biological and personal factors interact with environmental factors to affect risk for CMD in the real world and across the life course [112]. While we only examined one environmental factor herein (PM_2.5_ exposure) and only across a limited time window, future studies accounting for the totality of exogenous (external) and endogenous (internal) exposures from conception onwards, and across generations, offer promise to simultaneously distinguish, characterize and quantify etiologic, mediating, moderating, and co-occurring risk and protective factors and their relationship to the onset, progression, and outcomes of personal health and population-level disparities [112]. We plan to expand our analysis in future research that incorporates over 20,000 environmental exposures and daily measures of heat over 15 years linked to SCCS participant data which include five survey waves, Medicaid and Medicare (Parts A, B, C, and D) claims data, national death index files, state cancer registries for persons with cancer, and biological samples. This will allow us to apply an exposome-wide approach and examine complete exposure pathways from source of exposure, biomarkers of exposure, effect and disease susceptibility, to population-level disparities.

### Limitations

Study limitations include CMD was self-reported in response to a set of questions that asked the participant if s/he was ever told by their doctor that they had CVD, diabetes, etc. Self-reported health outcomes have been found to be affected by measurement error as the result of recall bias and social desirability [113]. However, some studies have found good concordance between self-report and more objective measures, such as medical records to identify disease history [113]. In addition, while the 3 km gridded measure of PM_2.5_ is a used as a proxy for exposure at the level of residential address, it is superior to either ambient or satellite measures used alone, which is a typical course of action.

## Figures and Tables

**Figure 1 ijerph-17-03561-f001:**
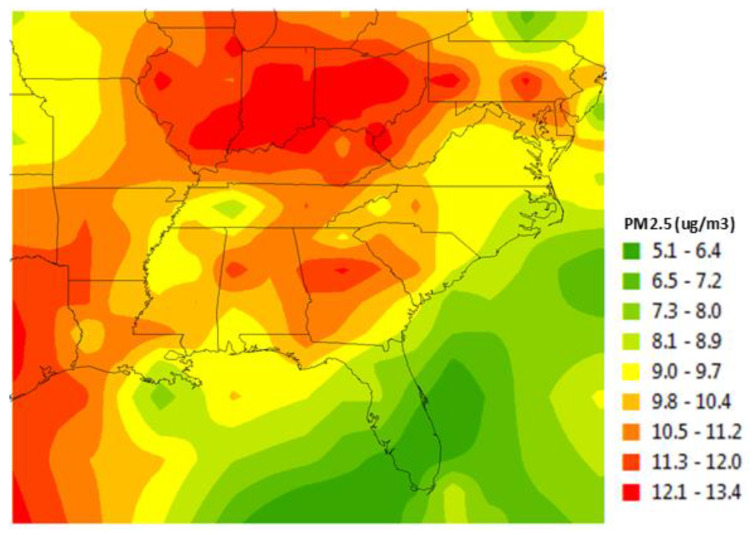
Mean annual 3-km fine particulate matter (PM_2.5_) for 2009.

**Figure 2 ijerph-17-03561-f002:**
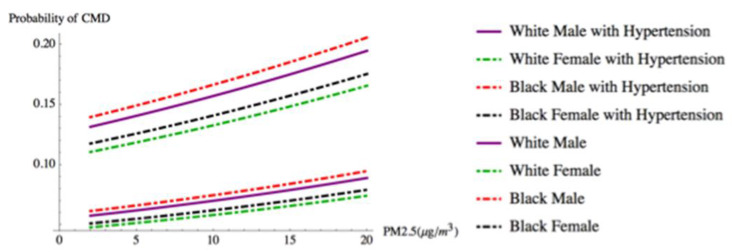
Individuals by race and gender with and without hypertension.

**Figure 3 ijerph-17-03561-f003:**
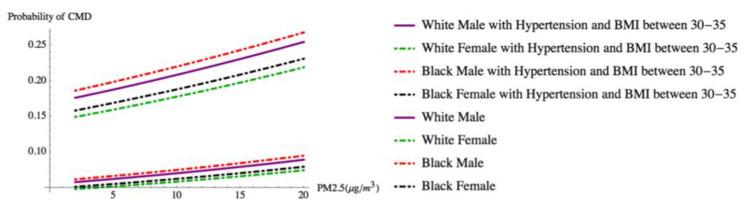
Individuals by race and gender with hypertension and BMI between 30 and 35 vs. individuals by race and gender without hypertension and BMI below 30.

**Table 1 ijerph-17-03561-t001:** Environmental risk factors for cardio-metabolic diseases.

Domain	Subdomain	Environmental Stressor	Cardiovascular Disease (CVD)	Stroke	Diabetes
Natural	Metals	Lead	Cosselman [58]	Navas [59]	Orioli [60]
		Arsenic	Smith [61]	Smith [61]	Smith [61]
			Cosselman [58]		
		Cadmium	Cosselman [58]	Peters [62]	Edwards [63]
	Solvents and pesticides	Solvents	Bulka [64]	Rinsky [65]	Montgomery [66]
		Pesticides	Wilcosky [67]		
	Air pollution	PM10, PM2.5, ultrafine PM	Mohammadi, [68]	Kowalska [69]	
	Gases	Carbon monoxide	Lee [70]	Hampson [71]	Huang [72]
		Ozone	Goodman [73]	Srebot [74]	Jerrett [75]
		Nitrogen dioxide Sulfur dioxide	Kopp [76]	Amancio [77]	Coogan [40]
Built	Neighborhood conditions	Walkability	Gaglioti [78]	Kwon [79]	Sundquist [80]
		Perceived/actual safety			Pham [81]
					Evonson [82]
	Access to healthy foods	Availability of healthy or unhealthy stores/restaurants	Lindberg [83]		Christine [84]
			Gaglioti [78]		
			Peolman [85]		
Social	Demographic	Population density	Rodriguez [86]		
		Socioeconomic status (SES)	Gebreab [87]		
		Social supports		Brown [88]	Zhang [50]
	Access to health care	Access to insurance and health care services	Li [89]	Medford-Davis [90]	Stark [91]
	Social stressors	Community stressors Residential segregation	Ford [91]	Booth [92]	Grigsby-Toussaint [93]
			Kershaw [94]	Patel [95]	
	Cultural influences	Lack of trust in health care providers	Schoenthaler [96]		Heisler [97]
		Socio-cultural beliefs and norms			
Policy		Dietary policy	Pearson [98]	Jilcott Pitts [99]	
		Physical activity policy		Jilcott Pitts [99]	
		Endocrine-disrupting chemicals policies			Shaikh [100]
		diabetes care and prevention policy			Ackermann [101]

**Table 2 ijerph-17-03561-t002:** Characteristics of the sub-cohort of Southern Community Cohort Study (*n* = 48,799).

Characteristics	% of Sample	% with CMD	Sig.
**All Participants**	100.0	29.2	
Gender			<0.001
Male	39.6	27.6	
Female	60.4	30.3	
**Race**			0.072
**Black**	66.1	28.9	
Male (*n* = 13,292)	27.2	25.9	
Female (*n* = 18,986)	38.9	31.0	
**White**	33.9	29.7	
Male (*n* = 6046)	12.4	31.1	
Female (*n* = 10,475)	21.5	28.9	
**Education (years completed)**			0.001
Less than 9 years	7.5	41.7	
9–11 years	21.0	32.4	
12 years (or GED)	34.3	27.8	
Vocational/technical	5.0	29.9	
Some college	20.1	27.0	
College graduate	7.7	24.7	
Graduate school	3.2	21.4	
Doctorate	1.3	19.5	
**Marital Status**			0.001
Married or with partner	34.3	29.6	
Divorced	34.4	29.1	
Widowed	9.4	41.6	
Single	21.9	23.5	
**Household Income**			0.001
<$15,000	56.4	32.1	
$15,000–$24,999	21.1	27.6	
$25,000–$49,999	13.9	25.7	
$50,000–$99,999	6.6	21.1	
>$100,000	2.0	14.5	
**Residence**			0.001
Urban	54.2	26.4	
Rural	45.8	32.5	
**Air Quality Inside**			0.105
Poor	5.8	28.3	
Fair	28.5	28.7	
Good	52.7	29.4	
Excellent	12.9	30.1	
**Air Quality Outside**			0.001
Poor	7.2	31.5	
Fair	34.4	28.3	
Good	46.3	29.6	
Excellent	12.1	28.8	
**Body Mass Index (BMI)**			0.001
Less than or equal 18.5	1.3	16.8	
18.5–25	24.0	17.2	
25–30	29.5	25.3	
30–35	21.9	34.0	
35–40	12.0	40.5	
40 or higher	11.2	45.3	
**Employment Status**			0.001
**Employed**	38.5	19.9	
Not employed	61.5	35.0	
**Age**			0.001
Senior 65 years and older	10.3	45.2	
40–64 years old	89.7	27.3	
**Smoking Status**			0.001
Current	42.0	23.9	
Former	22.6	38.0	
Never	35.4	29.8	
**Hypercholesterolemia**			0.001
No	65.3	19.4	
Yes	34.7	47.6	
**Hypertension**			0.001
No	44.4	14.3	
Yes	55.6	41.1	

**Table 3 ijerph-17-03561-t003:** Generalized linear mixed model fixed effects estimates for presence of cardio-metabolic disease in FQHC cohort participants of the Southern Community Cohort Study (*n* = 48,799).

Model Term	Coefficient	Std. Error	*T*-Value	*p*-Value	95% Confidence Interval		Odds Ratio	95% Confidence Interval for Exp (Coefficient)	
					Lower	Upper		Lower	Upper
**Intercept**	−3.511	0.1482	−23.692	0	−3.802	−3.221	0.030	0.022	0.04
**Enrollment Age**	0.025	0.001	24.005	0	0.023	0.027	1.025	1.023	1.027
**Education**									
Doctorate	−0.2	0.1407	−1.419	0.156	−0.475	0.076	0.819	0.622	1.079
Masters	−0.282	0.0579	−4.867	0	−0.395	−0.168	0.754	0.673	0.845
College	−0.142	0.0638	−2.232	0.026	−0.267	−0.017	0.867	0.765	0.983
Some College	−0.128	0.0355	−3.607	0	−0.198	−0.059	0.880	0.820	0.943
Vocational	−0.034	0.0353	−0.953	0.341	−0.103	0.036	0.967	0.902	1.036
High School	−0.162	0.0286	−5.673	0	−0.218	−0.106	0.850	0.804	0.899
Some High School	−0.064	0.0488	−1.311	0.19	−0.16	0.032	0.938	0.852	1.032
Less 9 years education	REF	-	-	-	-	-	-	-	-
**Marital Status**									
Single	−0.109	0.038	−2.878	0.004	−0.184	−0.035	0.896	0.832	0.966
Widowed	0.068	0.037	1.835	0.067	−0.005	0.14	1.070	0.995	1.151
Divorced	−0.021	0.0287	−0.727	0.467	−0.077	0.035	0.979	0.926	1.036
Married	REF	-	-	-	-	-	-	-	-
**Income**									
$100,000 plus	−0.713	0.0677	−10.521	0	−0.845	−0.58	0.490	0.429	0.56
$50,000–$99,000	−0.381	0.06	−6.356	0	−0.499	−0.264	0.683	0.607	0.768
$25,000–$49,000	−0.199	0.0429	−4.632	0	−0.283	−0.115	0.820	0.754	0.892
$15,000–$24,000	−0.124	0.0233	−5.337	0	−0.170	−0.079	0.883	0.844	0.924
Less than $15,000	REF	-	-	-	-	-	-	-	-
**Rural or Farm**									
Rural or Farm	0.036	0.025	1.435	0.151	−0.013	0.085	1.037	0.987	1.089
Urban	REF	-	-	-	-	-	-	-	-
**Air Quality Outside**									
Excellent	−0.113	0.0384	−2.951	0.003	−0.189	−0.038	0.893	0.828	0.963
Good	−0.033	0.036	−0.921	0.357	−0.104	0.037	0.967	0.902	1.038
Fair	−0.061	0.0342	−1.777	0.076	−0.128	0.006	0.941	0.880	1.006
Poor	REF	-	-	-	-	-	-	-	-
**Air Quality Inside**									
Excellent	0.15	0.0515	2.913	0.004	0.049	0.251	1.162	1.050	1.285
Good	0.01	0.0458	0.218	0.827	−0.080	0.1	1.010	0.923	1.105
Fair	0.038	0.0424	0.888	0.374	−0.045	0.121	1.038	0.956	1.128
Poor	REF	-	-	-	-	-	-	-	-
**BMI**									
40 or higher	1.061	0.1134	9.351	0	0.838	1.283	2.888	2.313	3.607
35–39	0.824	0.1051	7.846	0	0.618	1.03	2.280	1.856	2.801
30–34	0.597	0.0937	6.374	0	0.413	0.781	1.817	1.512	2.183
25–29	0.252	0.1101	2.291	0.022	0.036	0.468	1.287	1.037	1.597
18.5–24	−0.015	0.1042	−0.143	0.886	−0.219	0.189	0.985	0.803	1.208
Less than 18.5	REF	-	-	-	-	-	-	-	-
**Hypertension**									
Yes	0.907	0.0284	31.983	0	0.852	0.963	2.478	2.344	2.620
No	REF	-	-	-	-	-	-	-	-
**Hypercholesterol**									
Yes	0.959	0.0187	51.155	0	0.922	0.996	2.609	2.515	2.706
No	REF	-	-	-	-	-	-	-	-
**Employment**									
Yes	−0.504	0.0222	−22.681	0	−0.548	−0.461	0.604	0.578	0.631
No	REF	-	-	-	-	-	-	-	-
**Race**									
Black	0.069	0.0478	1.435	0.151	−0.025	0.162	1.071	0.975	1.176
White	REF	-	-	-	-	-	-	-	-
**Smoking History**									
Never Smoked	−0.017	0.0344	−0.487	0.626	−0.084	0.051	0.983	0.919	1.052
Former Smoker	0.19	0.0296	6.404	0	0.132	0.248	1.209	1.141	1.281
Current Smoker	REF	-	-	-	-	-	-	-	-
**PM_2.5_**	0.026	0.0093	2.751	0.006	0.007	0.044	1.026	1.007	1.045
**Gender**									
Female	−0.197	0.022	−8.93	0	−0.24	−0.154	0.821	0.787	0.858
Male	REF	-	-	-	-	-	-	-	-

**Table 4 ijerph-17-03561-t004:** Random effect.

Random Effect Covariance	Estimate	Std. Error	Z	*p*-Value	95% Confidence Interval	
					Lower	Upper
Var (Intercept)	0.017	0.008	1.984	0.047	0.006	0.045

Covariance structure: variance components; subject specification: state.

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
