# Peer review of "The Effects of Social, Personal, and Behavioral Risk Factors and PM2.5 on Cardio-Metabolic Disparities in a Cohort of Community Health Center Patients"

_ijerph, 2020, doi:10.3390/ijerph17103561_

Round 1
Reviewer 1 Report
As a scientist who has also studied air quality, in which we collected PM2.5 and PM10, along with obtaining various health data from other ethnic groups in SE Asia, I can say, without question, that the methods, science, and quality of this article is exceptional. This article's research design is quite impressive and novel!
While there is virtually no area I can critique that has not been already considered, I did not necessarily see any citations that reflect the global south. Granted, this particularly study is EXCELLENT meritoriously in its own right, I would have, ideally, like to have seen a sprinkle or two of articles that reflect similar methods undertaken by other researchers doing similar (not the same) research globally.
Apart from these comments, this article is excellent and deserves to be published posthaste. Thank you for allowing me this opportunity to review.
Author Response
Reviewer 1:
As a scientist who has also studied air quality, in which we collected PM2.5 and PM10, along with obtaining various health data from other ethnic groups in SE Asia, I can say, without question, that the methods, science, and quality of this article is exceptional. This article's research design is quite impressive and novel!
While there is virtually no area I can critique that has not been already considered, I did not necessarily see any citations that reflect the global south. Granted, this particularly study is EXCELLENT meritoriously in its own right, I would have, ideally, like to have seen a sprinkle or two of articles that reflect similar methods undertaken by other researchers doing siilar (not the same) research globally.
Response: We cited two studies (Al-Hamdan, 2009, p. 5, line 153; Al-Hamdan, 2014, p. 11, line 260) that used similar methodology in a different study (REGARDS). While we were able to identify other studies that were undertaken in other counties that used satellite AOD data, we did not identify any that combined ground monitoring station data with satellite unable to find any other studies. The two Al-Hamdan studies identified above were the only studies that used a similar methodology.
Reviewer 2 Report
I have several concerns about this paper due to methods and how results are presented and interpreted.
Selection of cohort for analysis Dropped n=12,298 subjects with data unavailable for PM25, unclear how or why it is unavailable – are certain 3km areas not available and why? Dropped n=18,598 due to missing data in covariates for modeling – was there a particular covariate that was missing for a majority of subjects? Or is it missing across covariates that is the major culprit? Dropped another n=4,818 that were enrolled with a mailed survey versus through the community health center – justification for dropping these subjects? Case deletion assumes data are missing completely at random – no data is given on subjects that were deleted – and no discussion of the potential limitations of interpretation of results due to case deletion. Information on outcome and covariates for all subjects and how changes when subjects are case deleted is needed. Unit of clustering, no data is given on number of subjects per cluster/state. What was the estimated intraclass correlation coefficient? Would a unit of cluster at the county level be more appropriate? Justification for state as a cluster level is weak. No justification that covariance structure of compound symmetry is the best fit; seems plausible that subjects who live farther apart within a state may be less correlated than those that live close together. Were different covariate structure examined and the best fit covariate structure determined to be compound symmetry? Generalized linear models for a binary outcome are subject-specific models and parameters should be interpreted as such – fixed effect parameters are not population averages but conditional on random effect of 0. Can convert subject-specific parameter estimates to population averages. For categorical predictors, overall test results should be given Information on whether assumption of linearity for the functional form of PM25 is appropriate is needed Composite outcomes can be difficult to interpret. No data is given on distribution of individual components of the CMD outcome or if have the distribution - and how the individual distributions may vary by covariates. It seems like sensitivity analysis on the individual components may be useful. Results are very difficult to interpret and synthesize; they are given in combination of tables and text. Table 3 should have columns for yes and no for CMD as well as overall to make easier for reader to interpret. Things that are mentioned in text are not in the tables – full model table may be better to present in appendix and include unadjusted and adjusted odds ratios all in one table? Potential for unmeasured confounding should be a limitation. Software used for fitting models should be given.
Author Response
Reviewer 2:
Selection of cohort for analysis Dropped n=12,298 subjects with data unavailable for PM25, unclear how or why it is unavailable – are certain 3km areas not available and why? Dropped n=18,598 due to missing data in covariates for modeling – was there a particular covariate that was missing for a majority of subjects? Or is it missing across covariates that is the major culprit? Dropped another n=4,818 that were enrolled with a mailed survey versus through the community health center – justification for dropping these subjects?
Case deletion assumes data are missing completely at random – no data is given on subjects that were deleted – and no discussion of the potential limitations of interpretation of results due to case deletion. Information on outcome and covariates for all subjects and how changes when subjects are case deleted is needed. Unit of clustering, no data is given on number of subjects per cluster/state.
Response: All survey data were 100% complete. While we had complete daily PM2.5 exposure data for all ~85,000 participants, due to the rolling enrollment (2003-2009), not all of them had 12 months of daily PM2.5. We started with all (n= 84,513) SCCS participants but excluded those who: 1) did not have 12 months of daily PM2.5 data (n=12,299), 2) were not recruited through a community Health center (n=18,597), and 3) did not identify as African American or White (n= 4,818). The following exclusion criteria were used to derive the final sample.
SCCS Exclusion Critieria
Total Initial sample
84,513
Removed those who did not have 12 months of daily PM2.5 data 72,214
Removed those who were not recruited through a CHC 53,617
Removed those who did not identify as AA or white 48,799
Final sample 48,799
There were significant differences between those recruited through CHCs and those who were not. Those who were recruited though media and flyers were much more likely to be white, married, and to have a higher income. We decided not to include them because we wanted to ensure we were comparing cardio-metabolic disparities among African Americans and White who had similar demographic backgrounds.
The following information regarding the number of subjects per cluster/state and their corresponding percentages:
What was the estimated intraclass correlation coefficient? Would a unit of cluster at the county level be more appropriate? Justification for state as a cluster level is weak.
Response: The estimated intraclass correlation for the generalized linear mixed model with logit link is equal to 0.00514. We had limited access to geographic information due to PHI restrictions and only had access to state identifiers.
No justification that covariance structure of compound symmetry is the best fit; seems plausible that subjects who live farther apart within a state may be less correlated than those that live close together.
Were different covariate structure examined and the best fit covariate structure determined to be compound symmetry?
Response: Not applicable. We have only used random intercept in our modeling.
Generalized linear models for a binary outcome are subject-specific models and parameters should be interpreted as such – fixed effect parameters are not population averages but conditional on random effect of 0. Can convert subject-specific parameter estimates to population averages. For categorical predictors, overall test results should be given
Response: The overall model F-statistic =458.565 ; the P-value < 0.001.
Information on whether assumption of linearity for the functional form of PM25 is appropriate is needed
Response: The pattern shown in the following graph indicates that the linearity assumption for PM2.5 is satisfied
Composite outcomes can be difficult to interpret. No data is given on distribution of individual components of the CMD outcome or if have the distribution - and how the individual distributions may vary by covariates. It seems like sensitivity analysis on the individual components may be useful.
Response: Individual distribution is given below. In addition, the purpose of this research was to look at all three CMD components combined. We will do a further detailed analysis of each component in a future paper.
Results are very difficult to interpret and synthesize; they are given in combination of tables and text. Table 3 should have columns for yes and no for CMD as well as overall to make easier for reader to interpret.
Things that are mentioned in text are not in the tables – full model table may be better to present in appendix and include unadjusted and adjusted odds ratios all in one table?
Potential for unmeasured confounding should be a limitation. Software used for fitting models should be given.
Response We modified Table 3 to clarify p values. The following text was inserted on p. 7, lines 196-197.
“IBM SPSS 26 was used to perform statistical analysis and Mathematica software was used to draw graphs.”
Reviewer 3 Report
This study assessed the relationship between air pollution, measured using daily PM2.5, and cardio-metabolic disease (CMD) in a cross-sectional sample of participants from a cohort study in the Southeast United States (U.S.). Overall, this is an interesting study that provides novel public health data addressing the influence of environmental pollution on CMD. Particular strengths of this study include the use of a cohort from a health disparate area of the U.S., as well as the inclusion of multiple individual-level covariates that may be related to CMD and underlie potential health disparities. However, there are a few points of concern that this reviewer would like to highlight, which are outlined below for the authors. Addressing these issues will strengthen the overall quality of this study and its results.
Major Points:
In the abstract, the authors note that disparities between Blacks and Whites will be a focus of this study. However, there is no discussion of the published literature that addresses racial disparities in the relationship between CMD and air pollution. Further, in their statement of the objective of this study in the final paragraph of the Introduction, there is also no mention of Black/White health disparities being a focus of this study. In the fourth paragraph of the Introduction, the authors first mention PM, but do not provide a full description of this acronym until the Procedures section of the Methods. It would be beneficial to a more thorough description of PM in the Introduction to better understand that it is a form of air pollution.Minor Points:
The authors use both Black and African American interchangeably throughout the manuscript. It may be advisable to choose one or the other and use consistently.Author Response
In the abstract, the authors note that disparities between Blacks and Whites will be a focus of this study. However, there is no discussion of the published literature that addresses racial disparities in the relationship between CMD and air pollution.
Response: Inserted on p. 3 Line 102): Several studies have found black:white racial disparities in the association between exposure to PM2.5 and cardio-metobolic outcomes. The MESA study (Hicken, et al., 2016, Epidemiogy, 27(1): 42-50. doi:10.1097/EDE.0000000000000367.) found that Blacks compared to Whites, showed a stronger adjusted association between air pollution and left-ventricular mass index (LVMI) and left-ventricular ejection fraction (LVEF). The MESA study also found that higher exposures to multiple chemical constituents of air pollution may be a novel contributor to diabetes disparities (Ruiz D, Becerra M, Jagai JS, Ard K, Sargis RM. Disparities in Environmental Exposures to Endocrine-Disrupting Chemicals and Diabetes Risk in Vulnerable Populations. Diabetes Care. 2018;41(1):193. doi: 10.2337/dc16-2765).Data from the HeartSCORE study found significant black:white racial disparities between exposure to PM2.5 and higher blood glucose, worse arterial endothelial function, and incident CVD events (Erqou, et al, 2018, Particulate Matter Air Pollution and racial differences in cardiovascular disease risk, Atheriosclerosis, Thrombosis, and Vascular Biology, 38(4), 935-942.
Further, in their statement of the objective of this study in the final paragraph of the Introduction, there is also no mention of Black/White health disparities being a focus of this study.
Response: Inserted on p3, line 114: This study examined the effects of PM2.5 exposure on black:white disparities in CMD.
In the fourth paragraph of the Introduction, the authors first mention PM, but do not provide a full description of this acronym until the Procedures section of the Methods. It would be beneficial to a more thorough description of PM in the Introduction to better understand that it is a form of air pollution.
Response: Added on p. 2, lines 81-86: Exposures to toxicants in the natural environment that have been linked to CMD outcomes include heavy metals (lead, mercury, cadmium, and arsenic), solvents, pesticides, indoor pollution (second hand smoke, biomass fuels), outdoor air pollution comprised of complex mixtures of gases that include particulate matter (PM), which includes PM10 (course), PM2.5 (fine) and ultrafine PM, carbon monoxide [CO], ozone [O3], nitrogen dioxide [NO2], sulfur dioxide [SO2] and diesel and other sources of (see Table 2). PM is a mixture of solids and liquid droplets present in the air that vary in mass, number, size, shape, surface area, chemical composition as well as reactivity, acidity, solubility and origin [31].
Minor Points:
The authors use both Black and African American interchangeably throughout the manuscript. It may be advisable to choose one or the other and use consistently.
Response: This discrepancy was addressed. The term African Americans was dropped in lieu of Blacks.
Round 2
Reviewer 2 Report
Comments to Authors
I have the similar concerns as first review as no new data or information was given in the response.
- Selection of cohort for analysis
- Dropped n=12,298 subjects with data unavailable for PM25, unclear how or why it is unavailable – are certain 3km areas not available and why?
- Dropped n=18,598 due to missing data in covariates for modeling – was there a particular covariate that was missing for a majority of subjects? Or is it missing across covariates that is the major culprit?
- Dropped another n=4,818 that were enrolled with a mailed survey versus through the community health center – justification for dropping these subjects?
- Response from author:
- Removed those who did not have 12 months of daily PM2.5 data 72,214
- Removed those who were not recruited through a CHC 53,617
- Removed those who did not identify as AA or white 48,799
- Final sample 48,799
- Text from paper does not appear to match; “For the current study, participants were limited to those for whom 12 months of daily PM5 data at date of enrollment were available (n=72,215). Only persons who identified as either white or Black and who had complete data for variables included in the model were included (n=53,617). Persons who were not recruited through community health centers were excluded from the study. The final sample thus was comprised of those persons for whom we had 12 months of daily PM2.5 data, were either Black or white and had no missing data of those variables that were used in the model, and were recruited through community health centers (n=48,799).
- Case deletion assumes data are missing completely at random – no data is given on subjects that were deleted – and no discussion of the potential limitations of interpretation of results due to case deletion. Information on outcome and covariates for all subjects and how changes when subjects are case deleted is needed.
- Unit of clustering, no data is given on number of subjects per cluster/state.
- No justification that covariance structure of compound symmetry is the best fit; seems plausible that subjects who live farther apart within a state may be less correlated than those that live close together. Were different covariate structure examined and the best fit covariate structure determined to be compound symmetry?
- Response from author:
- Not applicable. We have only used random intercept in our modeling.
- Yes a random intercept only is an assumption of compound symmetry, this seems a limitation due to the spatial nature of the data that is not being accounted for or no justification has been given that this is an appropriate assumption. State that because of PHI only have state level data, but then not clear how map 3km PM5 data to subject if only have state level data? It does seem like some type of modeling accounting for spatial correlation structure would be appropriate.
- Generalized linear models for a binary outcome are subject-specific models and parameters should be interpreted as such – fixed effect parameters are not population averages but conditional on random effect of 0. Can convert subject-specific parameter estimates to population averages.
- Information on whether assumption of linearity for the functional form of PM25 is appropriate is needed
- Potential for unmeasured confounding should be a limitation.
Author Response
Please see the response attached.

Reviewer 3 Report
The authors have addressed each of the concerns raised previously. The manuscript is stronger for these efforts. Nicely done.
Author Response
Thank you for the positive feedback.